

# Compiled records of carbon isotopes in atmospheric CO₂ for historical simulations in CMIP6

Heather Graven[1], Colin E. Allison[2], David M. Etheridge[2], Samuel Hammer[3], Ralph F. Keeling[4], Ingeborg Levin[3], Harro A. J. Meijer[6], Mauro Rubino[5], Pieter P. Tans[7], Cathy M. Trudinger[2], Bruce H. Vaughn[8], James W. C. White[8]

[1]Department of Physics, Imperial College London, UK
[2]CSIRO Climate Science Centre, Oceans and Atmosphere, Aspendale, Australia
[3]Institut für Umweltphysik, Heidelberg University, Germany
[4]Scripps Institution of Oceanography, University of California, San Diego, USA
[5]Dipartimento di Matematica e Fisica, Università della Campania "Luigi Vanvitelli", Caserta, Italy
[6]Center for Isotope Research, University of Groningen, The Netherlands
[7]National Oceanic and Atmospheric Administration, Boulder, USA
[8]Institute of Arctic and Alpine Research, University of Colorado, Boulder, USA

*Correspondence to*: Heather Graven (h.graven@imperial.ac.uk)

**Abstract.** The isotopic composition of carbon ($\Delta^{14}C$ and $\delta^{13}C$) in atmospheric CO₂ and in oceanic and terrestrial carbon reservoirs is influenced by anthropogenic emissions and by natural carbon exchanges, which can respond to and drive changes in climate. Simulations of $^{14}C$ and $^{13}C$ in the ocean and terrestrial components of Earth System Models (ESMs) present opportunities for model evaluation and for investigation of carbon cycling, including anthropogenic CO₂ emissions and uptake. The use of carbon isotopes in novel evaluation of the ESMs' component ocean and terrestrial biosphere models and in new analyses of historical changes may improve predictions of future changes in the carbon cycle and climate system. We compile existing data to produce records of $\Delta^{14}C$ and $\delta^{13}C$ in atmospheric CO₂ for the historical period 1850-2015. The primary motivation for this compilation is to provide the atmospheric boundary condition for historical simulations in the Coupled Model Intercomparison Project 6 (CMIP6) for models simulating carbon isotopes in their ocean or terrestrial biosphere models. The data may also be useful for other carbon cycle modelling activities.

## 1 Introduction

The isotopic composition of carbon in atmospheric, ocean and terrestrial reservoirs has been strongly perturbed by human activities since the Industrial Revolution. Fossil fuel burning is diluting the proportion of the isotopes $^{14}C$ and $^{13}C$ relative to $^{12}C$ in atmospheric CO₂ by the addition of aged, plant-derived carbon that is partly depleted in $^{13}C$ and entirely depleted in $^{14}C$. This process is referred to as the Suess Effect following the early observations of radiocarbon in tree rings by Hans Suess (Suess, 1955;Revelle and Suess, 1957). The term Suess Effect was also later adopted for $^{13}C$ (Keeling, 1979). The magnitudes of the atmospheric $^{14}C$ and $^{13}C$ Suess Effects are determined not only by fossil fuel emissions, but also by carbon exchanges with the ocean and terrestrial reservoirs and the residence time of carbon in these reservoirs, which regulate the mixing of the fossil fuel signal out of the atmosphere (Stuiver and Quay, 1981;Keeling, 1979). In addition, some biological and physical processes cause isotopic fractionation and the associated fractionation factors can vary with environmental conditions. Land use changes can also influence carbon isotope composition (Scholze et al., 2008). Variations in $^{13}C$ are reported as $\delta^{13}C$, which represents deviations in $^{13}C/^{12}C$ from a standard reference



material (VPDB). For $^{14}$C, the notation $\Delta^{14}$C is used, which represents deviations from the Modern standard $^{14}$C/C ratio and includes a correction for mass-dependent isotopic fractionation based on $\delta^{13}$C as well as a correction for $^{14}$C radioactive decay of the sample (Stuiver and Polach, 1977).

In addition to the perturbation from fossil fuel emissions, atmospheric $\Delta^{14}$C was also subject to a large, abrupt perturbation in the 1950s and 60s when a large amount of $^{14}$C was produced during atmospheric nuclear

weapons testing. The introduction of this "bomb $^{14}$C" nearly doubled the amount of $^{14}$C in the Northern Hemisphere troposphere, where most of the tests took place (Rafter and Fergusson, 1957;Münnich and Vogel, 1958). Most testing stopped after 1962 due to the Partial Test Ban Treaty, after which tropospheric $\Delta^{14}$C decreased quasi-exponentially as bomb $^{14}$C mixed through the atmosphere and into carbon reservoirs in the ocean and terrestrial biosphere that exchange with the atmosphere on decadal timescales (Levin and

Hesshaimer, 2000).

Sustained, direct atmospheric measurements of the isotopic composition of $CO_2$ began in 1955 in Wellington, New Zealand for $\Delta^{14}$C, capturing the dramatic changes over the weapons testing period (Rafter and Fergusson, 1957;Manning et al., 1990;Currie et al., 2009). Observations of $\Delta^{14}$C at several more stations started in the late 1950s, with some ceasing operation by the 1970s (Nydal and Lövseth, 1983;Levin et al., 1985). For $\delta^{13}$C in

$CO_2$, sustained flask sampling programs began in 1977-78 at the South Pole (Antarctica), Christmas Island, and La Jolla and Mauna Loa (USA) (Keeling et al., 1979;Keeling et al., 2001), and in 1978 at Cape Grim (Australia) (Francey and Goodman, 1986;Francey et al., 1996). A global network for $\Delta^{14}$C measurements is currently run by Heidelberg University (Levin et al., 2010), while global networks for $\delta^{13}$C measurements are run by Scripps Institution of Oceanography (SIO) (Keeling et al., 2005), the Commonwealth Scientific and Industrial Research

Organisation (CSIRO) (Allison and Francey, 2007), and jointly by the University of Colorado Institute for Arctic and Alpine Research and the National Oceanic and Atmospheric Administration (referred to here as NOAA) (Vaughn et al., 2010). Several other groups are also conducting long-term isotopic $CO_2$ observations at individual sites or in regional networks.

Records of atmospheric $\Delta^{14}$C and $\delta^{13}$C have been extended into the past using measurements of cellulose from

tree rings and of $CO_2$ in air from ice sheets (ice cores and firn) respectively, recent examples include Reimer et al. (2013) and Rubino et al. (2013). Ice cores are generally not used to construct atmospheric $\Delta^{14}$C records due to in situ $^{14}$C production, and tree rings are generally not used to construct atmospheric $\delta^{13}$C records due to climatic and physiological influences on $^{13}$C discrimination. These records clearly show decreases in $\Delta^{14}$C and $\delta^{13}$C due to increased emissions of fossil-derived carbon following the Industrial Revolution and carbon from

land use change. Ice core, tree ring and other proxy records (e.g. lake macrofossil, marine foraminifera, coral and speleothem records) additionally reveal decadal to millennial variations associated with climate and carbon cycle variability, and, for $^{14}$C, changes in solar activity and the geomagnetic field (Damon et al., 1978).

Studies using $\Delta^{14}$C and $\delta^{13}$C observations of carbon in the atmosphere, ocean and terrestrial biosphere together with simulated $^{14}$C and $^{13}$C dynamics in models can provide insights to key processes in the global carbon cycle

including air-sea gas exchange, ocean mixing, water use efficiency in plants, and vegetation and soil carbon turnover rates. Ocean $\Delta^{14}$C observations have been separated into "natural" and "bomb" $^{14}$C components (Key et al., 2004) and combined with models to constrain the global air-sea gas exchange velocity (Naegler,



2009;Sweeney et al., 2007), and to constrain or identify biases in ocean model transport and mixing (Oeschger et al., 1975;Matsumoto et al., 2004;Khatiwala et al., 2009). Observations of $\delta^{13}C$ in ocean dissolved inorganic carbon have been used to investigate anthropogenic $CO_2$ uptake (Quay et al., 2003) and to evaluate ocean models that include marine ecosystem dynamics (Tagliabue and Bopp, 2008;Schmittner et al., 2013). With terrestrial biosphere models, simulations of the response of plants and photosynthesis to rising atmospheric $CO_2$ and changing water availability can be evaluated with $\delta^{13}C$ observations in atmospheric $CO_2$ or in leaves or tree rings, because a close relationship exists between processes controlling leaf-level isotopic discrimination and water-use efficiency (Randerson et al., 2002;Scholze et al., 2008;Ballantyne et al., 2011;Keller et al., 2017). Additionally, observations of $\Delta^{14}C$ can be used to constrain models of carbon turnover rates in vegetation and soil carbon at plot-level and global scales (Trumbore, 2000;Naegler and Levin, 2009).

The Coupled Model Intercomparison Project phase 6 (CMIP6, Eyring et al., 2016) is leading the coordination of current global earth system modelling activities. CMIP6 follows the previous phase CMIP5 that contributed to the Intergovernmental Panel on Climate Change's Fifth Assessment Report (IPCC, 2013). CMIP6 is organizing common standards for reporting of model output and protocols for a set of core experiments including historical simulations and for several additional specialized experiments. The specialized experiments focus on individual processes or time periods and they are referred to as CMIP6-endorsed MIPs, which are organized by separate committees. Ocean MIP (OMIP) focuses on historical ocean physics and biogeochemistry and provides a separate set of simulation protocols including climatic forcing provided by atmospheric reanalyses (Griffies et al., 2016;Orr et al., 2017). The Coupled Climate–Carbon Cycle Model Intercomparison Project (C4MIP) encompasses historical, future, and idealized biogeochemical simulations in both the ocean and the terrestrial biosphere, using climatic forcing from coupled ESMs as opposed to observations (Jones et al., 2016). For any historical simulations in CMIP6 that use observed atmospheric greenhouse gas concentrations to drive the Earth system models, compiled records of atmospheric $CO_2$ and other greenhouse gas concentrations are provided by Meinshausen et al. (2017). Here we describe a compilation of historical data for carbon isotopes in atmospheric $CO_2$ to support the inclusion of carbon isotope modelling in CMIP6. The carbon isotope datasets are provided in Table S1 and available at input4MIPs: https://esgf-node.llnl.gov/search/input4mips/.

By providing atmospheric datasets for $\Delta^{14}C$ and $\delta^{13}C$ in $CO_2$ as part of CMIP6, we hope to stimulate more activity in carbon isotope modelling. So far, the inclusion of carbon isotopes in large-scale models and model intercomparisons has been limited. Carbon isotopes were not included in CMIP5, the previous phase of coupled model intercomparison. One study, the Ocean Carbon Cycle Model Intercomparison Project 2 (OCMIP2), used simulations of ocean $^{14}C$ to evaluate modelled ocean circulation and its effects on simulated anthropogenic $CO_2$ uptake and marine biogeochemistry (Orr et al., 2001;Matsumoto et al., 2004). Model intercomparisons for $^{13}C$ in the ocean, and for both $^{13}C$ and $^{14}C$ in the terrestrial biosphere have not been performed. This may be partly a result of the small number of carbon cycle models presently simulating carbon isotopes, although some simulations with global models that don't explicitly include carbon isotopes have been possible by using off-line isotope models (Joos et al., 1996;Thompson and Randerson, 1999;Graven et al., 2012a;He et al., 2016).

In this paper, we first review how carbon isotopes are being included in the protocols for CMIP6, which are described in more detail in Orr et al. (2017) and Jones et al. (2016). We then describe the historical atmospheric



datasets we compiled for $\delta^{13}$C and $\Delta^{14}$C in $CO_2$.  We refer to the compiled datasets as "forcing datasets" to
emphasize that (i) they are intended for model input data, (ii) atmospheric observations have been used to
calculate annual and spatial averages, and (iii) some observations have been adjusted as described below.
Because of these modifications, the forcing data are not intended to be used in atmospheric inversions. For that
purpose we direct modellers to the original atmospheric observations used to produce the forcing datasets,

available through data repositories listed in Table 1. The firn and ice core data, updated from Rubino et al.
(2013), are available in Table S2.

## 2 Historical Simulations of Carbon Isotopes in CMIP6

For CMIP6, carbon isotopes are included in historical biogeochemical simulations as part of OMIP (Ocean MIP
(Orr et al., 2017)) and C4MIP (Coupled Climate–Carbon Cycle Model Intercomparison Project (Jones et al.,

2016)). Carbon isotopes will also be included in the simulation of future scenarios. In a separate paper, we will
provide atmospheric $\Delta^{14}$C and $\delta^{13}$C for future scenarios in CMIP6 created with a simple carbon cycle model
(Graven, 2015) and $CO_2$ emission and concentration scenarios from ScenarioMIP (O'Neill et al., 2016).

The CMIP6 simulation protocols for carbon isotopes are provided in detail in Orr et al. (2017) and Jones et al.
(2016), so only a short summary is given here. The variables requested for CMIP6 are stocks and fluxes of $^{14}$C

and $^{13}$C from any model including $^{14}$C or $^{13}$C in the land or ocean component. Stocks and fluxes of $^{14}$C should be
reported with a normalization factor of $1/Rs$ where $Rs$ is the standard $^{14}$C/C ratio, $1.176 \times 10^{-12}$ (Karlen et al.,
1965) whereas $^{13}$C should be reported without normalization. For the ocean, the variables requested are the net
air-sea fluxes of $^{14}$C and $^{13}$C and the dissolved inorganic $^{14}$C and $^{13}$C concentration (Jones et al., 2016;Orr et al.,
2017). Models simulating dissolved inorganic $^{13}$C concentration typically include $^{13}$C in their marine ecosystem

model (Tagliabue and Bopp, 2008;Schmittner et al., 2013) because the oceanic $\delta^{13}$C distribution is strongly
affected by marine productivity and organic matter remineralization. Models can simulate $^{14}$C as an abiotic
variable with corresponding abiotic carbonate chemistry (Orr et al., 2017) because the oceanic $\Delta^{14}$C distribution
is largely insensitive to biological activity (Fiadeiro, 1982), although some models might also include $^{14}$C in
their marine ecosystem model (Jahn et al., 2015). For the terrestrial biosphere, $^{14}$C and $^{13}$C fluxes associated

with gross primary productivity, autotrophic respiration and heterotrophic respiration are requested. Stocks of
$^{14}$C and $^{13}$C in vegetation, litter and soil should also be reported.

Expected uses for historical carbon isotope simulations in CMIP6 include the evaluation of modelled ocean $CO_2$
uptake and transport and carbon cycling in marine ecosystems, the evaluation of modelled carbon fluxes and
stocks in terrestrial ecosystems and the ecosystem responses to higher $CO_2$ and ecohydrological changes, and

the interpretation of atmospheric data. Including carbon isotopes in CMIP6 may also prepare for and motivate
more activity in carbon isotope modelling in future work.

## 3 Historical atmospheric forcing dataset for $\Delta^{14}$C in $CO_2$

We compiled historical data for $\Delta^{14}$C in $CO_2$ from tree ring records and atmospheric measurements to produce
the historical atmospheric forcing dataset. We use the data to estimate annual mean values for three zonal bands

representing the Northern Hemisphere (north of 30°N), the Tropics (30°S-30°N) and the Southern Hemisphere
(south of 30°S), shown in Figure 1.





For 1850-1940, we use estimates of $\Delta^{14}$C in $CO_2$ from previous compilations of tree rings and other records that define the calibration curves used for radiocarbon dating. Separate estimates have been made for the Northern Hemisphere, IntCal13 (Reimer et al., 2013), and for the Southern Hemisphere, SHCal13 (Hogg et al., 2013). Linear interpolation was used to estimate annual values from data with 5-year resolution provided by IntCal13 and SHCal13. We estimate $\Delta^{14}$C in the Tropics as the average of the Northern and Southern Hemispheres for

1850-1940. This estimate is consistent with annual tree ring measurements from 22°S in Brazil over 1927-1940 (Santos et al., 2015), which are 2.2 ± 2.5 ‰ lower than the average of Northern and Southern Hemisphere $\Delta^{14}$C. We did not find tropical tree ring data available for the period 1850-1927, but measurements from northern Thailand in an earlier period 1600-1800 were bracketed by measurements from New Zealand and USA (Hua et al., 2004), suggesting the average of Northern and Southern Hemisphere $\Delta^{14}$C is likely to provide a reasonable

estimate of tropical $\Delta^{14}$C.

For the period between 1940 and 1954, we set $\Delta^{14}$C in the Southern Hemisphere and Tropics to be the same as the Northern Hemisphere $\Delta^{14}$C, where Northern Hemisphere $\Delta^{14}$C is given by IntCal13 over 1940-50 and by tree ring data from Stuiver and Quay (1981) over 1951-54. We therefore fix the spatial $\Delta^{14}$C gradients at 0 ‰ over this period 1940-55. This approach is motivated by differences between the Southern Hemisphere $\Delta^{14}$C in

1950 in SHCal13 and another compilation of tree ring data by Hua et al. (2013). In SHCal13, Southern Hemisphere $\Delta^{14}$C becomes 4 ‰ higher than Northern Hemisphere $\Delta^{14}$C over 1940-50, after being similar to Northern Hemisphere $\Delta^{14}$C over 1915-40. However in Hua et al. (2013), Southern Hemisphere $\Delta^{14}$C is only 1 ‰ higher than Northern Hemisphere $\Delta^{14}$C, north of 40°N, in 1950. Tree ring measurements from Brazil in 1940-1954 (Santos et al., 2015) are also consistent with Northern Hemisphere $\Delta^{14}$C, with an average difference of -0.5

± 1.9 ‰ for the tree ring data minus Northern Hemisphere $\Delta^{14}$C in IntCal13. There is no significant difference between Northern Hemisphere $\Delta^{14}$C in IntCal13 and Hua et al. (2013) in 1950.

For Southern Hemisphere $\Delta^{14}$C over 1955-2015, we use direct measurements of atmospheric $\Delta^{14}$C, primarily the measurements conducted by Heidelberg University (Levin et al., 2010). However, we use data from 1955 through 1983 made by the Rafter Radiocarbon Laboratory at Wellington, New Zealand, to specify $\Delta^{14}$C in the

Southern Hemisphere for 1955-83. A correction of -4 ‰ is added to the Wellington data, as reported in Manning and Melhuish (1994), to account for a systematic difference between the Wellington and Heidelberg laboratories. For 1984-2014, data from Heidelberg University (Levin et al., 2010) from Neumayer (Antarctica), Cape Grim (Australia) and Macquarie Island (Australia) are averaged, if available, and where there is missing data the following procedure is used. If Macquarie Island data are missing, averages from Neumayer and Cape

Grim are adjusted by -1.2‰, the average difference in mean $\Delta^{14}$C across the three sites when available Macquarie data are included or not. This adjustment takes into account that $\Delta^{14}$C observed at Macquarie is lower than the stations further north and south, resulting from gas exchange over the Southern Ocean (Levin and Hesshaimer, 2000). Macquarie Island data are available for 1993-1999, 2003, 2007-09 and 2011. For 2015, only Neumayer data were available, so the annual mean at Neumayer was adjusted by -2‰, which is the mean

difference between Neumayer and the calculated Southern Hemisphere average for 2010-14.

For the Northern Hemisphere, $\Delta^{14}$C for 1955 to 1958 is based on tropospheric data compiled by Tans (1981). From 1959-1984, $\Delta^{14}$C observations from Vermunt, Austria are used (Levin et al., 1985). For the years 1985-86,



only a few observations from Vermunt are available. Observations by Heidelberg University from Izaña began in 1984 and from Jungfraujoch, Switzerland in 1986. Sampling at Alert, Canada started in 1989. We use data from Izaña for 1985-88, corrected for the mean difference between Izaña and the average of Izaña, Jungfraujoch, and Alert over 1989-97. For 1989-97, the average of Izaña, Jungfraujoch, and Alert is used. For 1997-2010, the average of Jungfraujoch, Alert and Mace Head, Ireland is used. From 2011-2015, only

Jungfraujoch data are available (Hammer and Levin, 2017), which were used here but adjusted by +0.4‰, taking into account that Jungfraujoch is influenced slightly more by fossil fuel $CO_2$ than Alert and Mace Head further north (see also Levin and Hesshaimer, 2000).

Observations in the tropics were made by the Heidelberg laboratory for the period 1991-97 at Mérida, Venezuela (8°N), and the annual averages at Mérida are used to specify tropical $\Delta^{14}C$ for 1991-97.

Measurements at Mérida were 2.9 ‰ higher than the Northern Hemisphere $\Delta^{14}C$ over 1991-97, on average, and the difference (2.9 ‰) was applied to Northern Hemisphere $\Delta^{14}C$ to estimate tropical $\Delta^{14}C$ for the periods 1985-1991 and 1998-2015. To estimate tropical $\Delta^{14}C$ before 1985, the atmospheric box model of V. Hesshaimer and T. Naegler with four tropospheric boxes separated at 30°N, Equator and 30°S (Naegler and Levin, 2006) was used together with Northern and Southern Hemisphere $\Delta^{14}C$ data. For each hemisphere, we calculated the ratio

between simulated annual $\Delta^{14}C$ averaged in the tropical boxes and simulated annual $\Delta^{14}C$ averaged in the polar boxes. The annual average tropical-to-polar ratio was then multiplied by the observed average Northern and Southern Hemisphere $\Delta^{14}C$ for each year to yield the values for the tropics.

A preliminary version of the $\Delta^{14}C$ data compilation (version 1) was released in early 2017 via email to C4MIP and OMIP researchers and at input4MIPs (https://esgf-node.llnl.gov/search/input4mips/). The version we

describe here (version 2) incorporates new and updated $\Delta^{14}C$ data from Heidelberg University, whereas version 1 was based on fewer data and on extrapolated values for the last few years. For the Northern Hemisphere, $\Delta^{14}C$ for 2011-15 was updated in version 2 but $\Delta^{14}C$ for 2010 and earlier is the same as in version 1. For the Southern Hemisphere, $\Delta^{14}C$ for 2000-2015 was updated. For the tropics, $\Delta^{14}C$ for 1998-2015 was updated. Differences between version 1 and version 2 are smaller than 3.5 ‰ for individual years, and average -0.2 ‰ for the

Northern Hemisphere (2011-2015), 0.0 ‰ for the tropics (1998-2015) and -1.3 ‰ for the Southern Hemisphere (2000-2015). Both versions are available at input4MIPs.

Here we make some comparisons with other reported atmospheric $\Delta^{14}CO_2$ measurements. Organized comparisons between laboratories conducting atmospheric $\Delta^{14}CO_2$ measurements using reference air were initiated around 2005, and results to date indicate that most laboratories are currently compatible within 2-3 ‰

(Miller et al., 2013;Hammer et al., 2016). In comparisons of the Southern Hemisphere $\Delta^{14}C$ forcing data with observations at the South Pole, differences are less than 2 ‰ with data from SIO and Lawrence Livermore National Laboratory (LLNL) over 2000-07 and differences are approx. 5 ‰ with University of Groningen data from 1987-89 (Meijer et al., 2006;Graven et al., 2012b). Observations from Wellington, New Zealand from 1983 to 2014 also show similar trends (Turnbull et al., 2016). Differences are less than 3.5 ‰ in comparisons of

the Northern Hemisphere $\Delta^{14}C$ forcing data with annual mean $\Delta^{14}C$ from SIO/LLNL observations at Point Barrow and La Jolla for 2002-07, and with University of Groningen observations at Point Barrow for 1987-89. The Northern Hemisphere $\Delta^{14}C$ forcing data also compares well with observations at Niwot Ridge, Colorado,



where trends of -4 to -6 ‰ yr$^{-1}$ have been observed since 2004 (Turnbull et al., 2007;Lehman et al., 2016).
Estimated tropical $\Delta^{14}$C shows good agreement with observations of $\Delta^{14}$C at Hawaii and Samoa for 2002-07
made by SIO/LLNL (Graven et al., 2012b), with differences less than 1.5 ‰ compared to the annual averages of
Mauna Loa, Kumukahi and Samoa. A limited amount of data available for Mauna Loa, Kumukahi and Samoa
from 2014-15 is also consistent with the estimated tropical $\Delta^{14}$C. Our estimate for tropical $\Delta^{14}$C in the 1960s lies

between observations from Ethiopia and Madagascar made at the Trondheim laboratory (Nydal and Lövseth,
1983). The Trondheim data were not used directly since no comparison between Heidelberg University and the
Trondheim laboratory took place, however future studies could potentially incorporate the Trondheim data,
which are available at the Carbon Dioxide Information Analysis Center (see Table 1).

A comparison with the data compilation covering 1950-2010 including tree ring and atmospheric $\Delta^{14}$C data by

Hua et al. (2013) is shown in Fig. 2. The range in tree ring $\Delta^{14}$C data overlaps the Northern and Southern
Hemisphere $\Delta^{14}$C forcing data in all periods, showing good consistency between the records. Hua et al. (2013)
separate data from tropical regions over the period 1950-1972, which also overlap the CMIP6 tropical $\Delta^{14}$C
forcing data for 1950-1972 (not shown in Fig. 2).

The data compilation shows the trends and spatial gradients in $\Delta^{14}$C of $CO_2$ over the historical period since

1850, as reported in previous studies. Between 1850 and 1952, atmospheric $\Delta^{14}$C decreased from approximately
-4 ‰ to a minimum value around -25 ‰ as emissions from fossil fuel combustion increased after the Industrial
Revolution. In the preindustrial and early industrial period to 1915, $\Delta^{14}$C was 3-6 ‰ lower in the Southern
Hemisphere than the Northern Hemisphere due to the negative influence of $CO_2$ exchange with aged, $^{14}$C-
depleted waters upwelling in the Southern Ocean (Braziunas et al., 1995;Rodgers et al., 2011;Lerman et al.,

1970;Levin et al., 1987). Between 1915 and 1955, the interhemispheric gradient decreased due to the growth in
fossil fuel emissions, which are concentrated in the Northern Hemisphere (McCormac et al., 1998). After 1955,
$\Delta^{14}$C increased rapidly as a result of nuclear weapons testing, reaching a maximum of 836 ‰ in the Northern
Hemisphere and 637 ‰ in the Southern Hemisphere, where the values 836 ‰ and 637 ‰ are the maxima in the
forcing data. $\Delta^{14}$C was even higher in the stratosphere and some Northern Hemisphere sites (Hesshaimer and

Levin, 2000). After 1963-64, tropospheric $\Delta^{14}$C decreased quasi-exponentially as the bomb $^{14}$C mixed with
oceanic and biospheric carbon reservoirs while growing fossil fuel emissions continued to dilute atmospheric
$^{14}CO_2$. Differences between the Northern and Southern Hemisphere reduced rapidly and were close to zero for
the 1980s-90s (Meijer et al., 2006;Levin et al., 2010;Levin and Hesshaimer, 2000), until the mid-2000s when a
Northern deficit in $\Delta^{14}$C emerged (Levin et al., 2010;Graven et al., 2012b). The Northern deficit in $\Delta^{14}$C has

been linked to a growing dominance of fossil fuel emissions in the Northern Hemisphere as air-sea exchange
with $^{14}$C-depleted ocean water in the Southern Hemisphere weakened with decreasing atmospheric $\Delta^{14}$C (Levin
et al., 2010;Graven et al., 2012b). $\Delta^{14}$C in background air has exhibited an average trend of about -5 ‰ yr$^{-1}$
since the 1990s (Graven et al., 2012c;Levin et al., 2013) and $\Delta^{14}$C in background air was 14-20 ‰ in 2015.

The CMIP6 forcing data for $\Delta^{14}$C are similar to the forcing data used in the ocean carbon cycle model

intercomparison OCMIP2 (Fig. 2), with a few notable differences. The zonal bands are defined slightly
differently: for CMIP6 we use boundaries of 30°N and 30°S whereas OCMIP2 used 20°N and 20°S. The
OCMIP2 forcing data include spatial differences over the period 1955-1968 only, for all other periods the same



$\Delta^{14}$C is used for all three zonal bands. Over the peak $\Delta^{14}$C period 1962-64 the OCMIP2 forcing data for the
tropics are 50-125 ‰ higher than CMIP6, whereas differences for the Northern and Southern Hemispheres are
smaller, less than 50 ‰. Larger differences in $\Delta^{14}$C in the tropics may be partly due to a lack of data in the
southern tropics over 1962-64. Measurements by Nydal and Lövseth (1983) in Madagascar (21°S) only started
in late 1964 and, compared to their observation sites in the northern tropics, the data from Madagascar show
tropical gradients up to 100‰ in 1965.

Two other periods where the CMIP6 forcing data noticeably deviate from the OCMIP2 forcing data are 1976-
82, when the OCMIP2 forcing data are 10-30 ‰ higher than the CMIP6 forcing data, and 1992-95, the OCMIP2
forcing data are approx. 10 ‰ lower. For these periods, the OCMIP2 forcing data appears to also be inconsistent
with the Hua et al. (2013) compilation (Fig. 2). The OCMIP2 forcing data ends in 1995, but the 1995 value (107
‰) is also appended for the year 2000 in the OCMIP2 forcing data, which is approx. 15 ‰ higher than the
observed $\Delta^{14}$C in 2000.

**4 Historical atmospheric forcing dataset for $\delta^{13}$C in CO$_2$**

We compiled historical data for $\delta^{13}$C in atmospheric CO$_2$ from ice core and firn records and from flask
measurements to produce the historical atmospheric forcing dataset. We use the data to estimate annual mean,
global mean values for $\delta^{13}$C (Fig. 1, Fig. 3) as described below. We combine the most extensive dataset for firn
and ice core $\delta^{13}$C measurements for the period after 1850 (Rubino et al., 2013) with recent flask data from three
laboratories, including the earliest flask data currently available. This approach provides a consistent data set
that incorporates ice core data spanning the historical period and higher temporal resolution in the flask data
available starting in 1978-80.

The ice core and firn $\delta^{13}$C records are from Law Dome and South Pole, measured by CSIRO (Rubino et al.,
2013). Due to the high snow accumulation rate at Law Dome, the firn and ice core record has high time
resolution, air age distribution 68% width of 8 years or less (Rubino et al., 2016), and overlaps the atmospheric
record. The datasets published in Rubino et al. (2013) were updated to the latest calibration scale at CSIRO in
September 2016. The calibration scale is revised from that presented in Allison & Francey (2007) with updated
corrections for cross contamination effects in the isotope ratio mass spectrometer ion source and ion correction
procedures for $^{17}$O interference (Brand et al., 2010). The calibration procedure also uses a link to the VPDB
reference scale established by the World Meteorological Organization Central Calibration Laboratory (CCL) for
stable isotopes in CO$_2$ (Max Planck Institute for Biogeochemistry, Jena, Germany). The revised procedure
ensures that all corrections are consistently applied to all samples measured at CSIRO since 1990, including all
ice-core, firn air and flask measurements. We do not include measurements conducted at NOAA reported in
Rubino et al. (2013).

We use atmospheric $\delta^{13}$C measured by flask sampling from CSIRO (Allison and Francey, 2007), NOAA
(Vaughn et al., 2004) and SIO (Keeling et al., 2001). Observations from SIO between 1977 and 1992 were made
in collaboration with the University of Groningen, using the analytical facilities at the University of Groningen
for $\delta^{13}$C measurement. After 1992, the SIO measurements were conducted solely at SIO (Guenther et al., 2001).
Observations of $\delta^{13}$C by CSIRO began at Cape Grim in 1978 and expanded to a global network in the 1980s.



CSIRO $\delta^{13}$C data prior to the early 1990s is not as well calibrated and therefore not publicly available except for Cape Grim, which has data available from the early 1980s. Observations of $\delta^{13}$C by NOAA and INSTAAR started in the early 1990s. Atmospheric $\delta^{13}$C data were downloaded in July 2016 from CSIRO, NOAA and SIO. Websites for data access are listed in Table 1.

       Here we use atmospheric $\delta^{13}$C data from two sites, South Pole (SPO) and Mauna Loa (MLO), in order to

capture interhemispheric differences in $\delta^{13}$C in defining a global mean value. These two sites are measured by all three laboratories. In order to compile data from the three laboratories, we used a third station, Alert, Canada, to assess inter-laboratory offsets, also referred to as "scale offsets". There is ongoing work in the community to incorporate best practices for preparing and measuring reference materials, and to use CCL $CO_2$-in-air reference materials to evaluate and resolve scale offsets in atmospheric $\delta^{13}$C data (Wendeberg et al., 2013;WMO/IAEA,

2016), however, not all currently reported data consistently accounts for any scale offsets between laboratories. We adjust SIO and NOAA $\delta^{13}$C data to be consistent with CSIRO $\delta^{13}$C data using scale offsets identified in measurements from Alert. Average differences between annual mean $\delta^{13}$C observations made at Alert for 2005-2015 by CSIRO and by the Max Planck Institute for Biogeochemistry (the CCL) are less than 0.01 ‰ (WMO/IAEA, 2016), so the compiled $\delta^{13}$C record we present here can also be regarded to be consistent with

the VPDB scale established by the CCL.

       Comparing annual mean $\delta^{13}$C observed at Alert over 1992-2014 shows that data reported by NOAA were 0.031 ‰ higher than CSIRO, and data reported by SIO were 0.046 ‰ higher than CSIRO, on average. The standard deviation in NOAA-CSIRO differences was 0.018 ‰, and 0.020 ‰ for SIO-CSIRO differences, with standard error of 0.004 ‰ for both. Similar offsets were found in comparisons of monthly mean rather than annual mean

$\delta^{13}$C at Alert, with larger standard deviations of 0.031 ‰ for monthly NOAA-CSIRO differences and 0.044 ‰ for monthly SIO-CSIRO differences, but smaller standard errors of 0.002 ‰ for both. The offset 0.031 ‰ was subtracted from NOAA data at South Pole and Mauna Loa, and 0.046 ‰ was subtracted from SIO data at South Pole and Mauna Loa. Then the monthly values from the three laboratories were averaged, and used to calculate combined annual means. As a result of varying data availability, data for 1977-90 at SPO and 1980-89 at MLO

is based only on SIO (with the offset applied), and data for 2015 is based only on CSIRO.

       The atmospheric data show a gradient of -0.043 ‰ between Mauna Loa and South Pole in 1980-84, growing to -0.095 ‰ in 2010-14 (Fig 3). Keeling et al. (2011) suggest that the preindustrial Northern Hemisphere – Southern Hemisphere $\delta^{13}$C gradient was +0.09 ‰, opposite in sign to the present interhemispheric gradient, using a regression of SIO $\delta^{13}$C data with global total fossil fuel emissions. They further demonstrate that the

inferred preindustrial gradient is consistent with a model of spatial variation in equilibrium fractionation during air-sea gas exchange. A similar preindustrial atmospheric $\delta^{13}$C gradient was simulated by Murnane and Sarmiento (2000) using a global ocean model, where they also attributed the primary driver of the gradient to equilibrium fractionation. $\delta^{13}$C data from Greenland ice cores and possibly deep firn are compromised by in situ $CO_2$ production so it is not possible to discern a precise preindustrial or pre-1980 $\delta^{13}$C gradient directly from

observations (Anklin et al., 1995;Francey et al., 1999;Tschumi and Stauffer, 2000;Jenk et al., 2016). We account for the possibility that $\delta^{13}$C measured in ice core and firn in Antarctica is slightly different from the global mean by using the regression from Keeling et al. (2011) to estimate $\delta^{13}$C at MLO. Then to estimate global $\delta^{13}$C we



average the observed Antarctic ice core and firn $\delta^{13}C$ and the estimated $\delta^{13}C$ for MLO. Previous studies have adjusted Antarctic ice core and firn $\delta^{13}C$ to estimate global levels by assuming that the preindustrial gradient was zero (Rubino et al., 2013).

To calculate a smoothed, global average time series for $\delta^{13}C$ in $CO_2$ over 1850-2015, we first average replicate measurements of ice core and firn samples from Rubino et al. (2013) and then calculate annual averages for any

year that includes an ice core or firn measurement. Using the annual averages, we estimate a corresponding $\delta^{13}C$ at MLO from the ice core and firn data using the regression from Keeling et al. (2011). We then append the ice core and firn data before 1977 to the annual average record at SPO beginning in 1978, omitting any ice core and firn data after 1978. We similarly append the $\delta^{13}C$ at MLO estimated from the ice core and firn data to the annual average record at MLO beginning in 1980, omitting any ice core and firn- based estimates after 1980

(Fig. 3). Smoothed curves were then calculated for SPO and MLO with stronger weighting on the recent flask-based data to account for the coarser time resolution of the ice core versus flask data due to diffusive smoothing in the firn. Then these curves were averaged and evaluated in the middle of each year 1850-2015 to produce the atmospheric forcing data (Fig 1).

Ice core and firn data after 1977 or 1980 were not used directly to produce the forcing data, but they are

included in Fig. 3 for comparison. The differences between SPO annual means and ice core and firn data after 1977 are less than 0.05 ‰ for 14 samples, and less than 0.09 ‰ for the two other samples. The differences between MLO annual means and estimates of MLO $\delta^{13}C$ from ice core and firn data after 1980 are less than 0.03 ‰ for 8 samples, and less than 0.09 ‰ for the one other sample. In addition, applying the regression to the annual mean SPO $\delta^{13}C$ flask data is consistent with MLO observations to within 0.05 ‰, suggesting the

regression from Keeling et al. (2011) using 1979-2003 SIO data is also consistent with combined means including the NOAA and CSIRO data, and with the longer period encompassing 2004-2015.

We also compare the global $\delta^{13}C$ forcing data with the global monthly mean Marine Boundary Layer $\delta^{13}C$ estimated from NOAA's larger network of stations, which is available for 1993-2015 (Fig. 3, www.esrl.noaa.gov/gmd/ccgg/mbl/mbl.html) (Masarie and Tans, 1995). Here the NOAA-CSIRO offset has

been applied (0.031 ‰). The differences for 1993-96 are -0.04±0.01 ‰, when the NOAA Marine BL global mean is close to MLO. Differences after 1996 are smaller, -0.02±0.01 ‰. Slightly lower values in NOAA Marine BL global mean indicates low- $\delta^{13}C$ air from the Northern Hemisphere is slightly underrepresented in the MLO-SPO average. However, the long-term trends are similar: both decreased by 0.6 ‰ between 1993 and 2015.

**5 Discussion and Conclusions**

We have produced a compilation of atmospheric datasets for $\Delta^{14}C$ and $\delta^{13}C$ in $CO_2$ over the historical period 1850-2015 with the aim of providing a standard atmospheric boundary condition for ocean and terrestrial biosphere models simulating [14]C and [13]C in CMIP6. The data can be accessed in in Table S1 and at input4MIPs: https://esgf-node.llnl.gov/search/input4mips/.



In compiling these atmospheric forcing datasets for $\Delta^{14}C$ and $\delta^{13}C$ in $CO_2$, our primary objective was to
        accurately and consistently compile the data available. We also aimed to provide datasets that are simple to use,
        particularly as $\delta^{13}C$ has not been included previously in a large model intercomparison. $\Delta^{14}C$ was included
        previously in OCMIP2 using a similar approach and atmospheric forcing dataset (Orr et al., 1999), but has been
        updated and improved in this version.

Several applications for simulations of $^{14}C$ and $^{13}C$ may require or benefit from the use of oceanic, terrestrial
        and/or other atmospheric data, including atmospheric data with higher temporal or spatial resolution. Available
        global-scale databases are listed in Table 1, and we encourage modellers to collaborate with data providers on
        model-data integration studies. In particular, data users should take care to account for any $\delta^{13}C$ scale offsets
        between laboratories, as described above.

We note that ice core and firn $\delta^{13}C$ data updated from Rubino et al. (2013) that were used to produce the $\delta^{13}C$
        forcing data are included in Table S2. The ice core and firn $CO_2$ data are also included in Table S2. These data
        could be joined with CSIRO observations at SPO, available from the World Data Centre for Greenhouse Gases
        (Table 1), to provide Antarctic records of $CO_2$ concentration and $\delta^{13}C$ measured in the same air sample by the
        same laboratory, which may be advantageous for some applications.

*Data and code availability.* The atmospheric forcing datasets for $\Delta^{14}C$ and $\delta^{13}C$ in $CO_2$ can be accessed in Table
        S1 and at input4MIPs: https://esgf-node.llnl.gov/search/input4mips/. Original atmospheric data are available
        from the websites listed in Table 1, and ice core and firn $\delta^{13}C$ data updated from Rubino et al. (2013) are
        included in Table S2. Interpolation and smoothing were conducted with standard routines in Matlab; further
        details are available from the lead author on request.

*Acknowledgments.* We thank the staff of the atmospheric monitoring stations for their long-term commitment to
        the flask sampling activities. The teams of the Law Dome and South Pole drilling expeditions provided the firn
        air and ice core samples. CSIRO GASLAB and ICELAB personnel supported the measurements of air from the
        CSIRO monitoring network and firn and ice core samples. Logistic support was provided by the Australian
        Antarctic Division (Macquarie Island, Law Dome) and the Bureau of Meteorology (Cape Grim, Macquarie
Island). The Australian Climate Change Science Program contributed to funding of the CSIRO measurements.
        Measurements at SIO were supported by the US National Science Foundation, Department of Energy, and
        NASA under grants 1304270, DE-SC0012167, and NNX17AE74G, by The Eric and Wendy Schmidt Fund for
        Strategic Innovation, and by NOAA for collection of air samples. Measurements at NOAA and INSTAAR were
        supported by the NOAA Climate Program Office. Measurements at Heidelberg University were partly funded
by a number of agencies in Germany and Europe, namely the Heidelberg Academy of Sciences, the Ministry of
        Education and Science, Baden-Wurttemberg, Germany; the German Science Foundation, the German Minister
        of Environment; the German Minister of Science and Technology; the German Umweltbundesamt and the
        European Commission, Brussels, as well as national funding agencies in Australia, Canada and Spain. H.
        Graven received support from the European Commission through a Marie Curie Career Integration Grant.






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



5    **Table 1. Available global-scale databases of $\Delta^{14}C$ and $\delta^{13}C$ in atmospheric $CO_2$, terrestrial and ocean carbon, and fossil fuel emissions.**

| Name | Type | Website |
|---|---|---|
| Scripps Institution of Oceanography Global $CO_2$ Program | $\Delta^{14}C$ and $\delta^{13}C$ in $CO_2$ | http://scrippsco2.ucsd.edu |
| NOAA Global Greenhouse Gas Reference Network | $\Delta^{14}C$ and $\delta^{13}C$ in $CO_2$ | https://www.esrl.noaa.gov/gmd/ccgg/ |
| World Data Centre for Greenhouse Gases (Including CSIRO data) | $\Delta^{14}C$ and $\delta^{13}C$ in $CO_2$ | http://ds.data.jma.go.jp/gmd/wdcgg/ |
| Heidelberg University data center | $\Delta^{14}C$ in $CO_2$ | https://heidata.uni-heidelberg.de/dataverse/carbon |
| Carbon Dioxide Information Analysis Center (CDIAC) | $\Delta^{14}C$ and $\delta^{13}C$ in $CO_2$, and $\delta^{13}C$ in fossil fuel $CO_2$ emissions | http://cdiac.ornl.gov/carbonisotopes.html http://cdiac.ornl.gov/trends/emis/meth_reg.html |
| GLobal Ocean Data Analysis Project GLODAP v2 | $\Delta^{14}C$ and $\delta^{13}C$ in ocean dissolved inorganic carbon | http://cdiac.ornl.gov/oceans/GLODAPv2/ |
| TRY Plant Trait Database | $\delta^{13}C$ in terrestrial plants | https://www.try-db.org/TryWeb/Home.php |
| International Tree-Ring Data Bank | $\delta^{13}C$ in terrestrial plants | https://www.ncdc.noaa.gov/data-access/paleoclimatology-data/datasets/tree-ring |
| Soil Carbon Database | $\Delta^{14}C$ and $\delta^{13}C$ in soil carbon | https://github.com/powellcenter-soilcarbon |




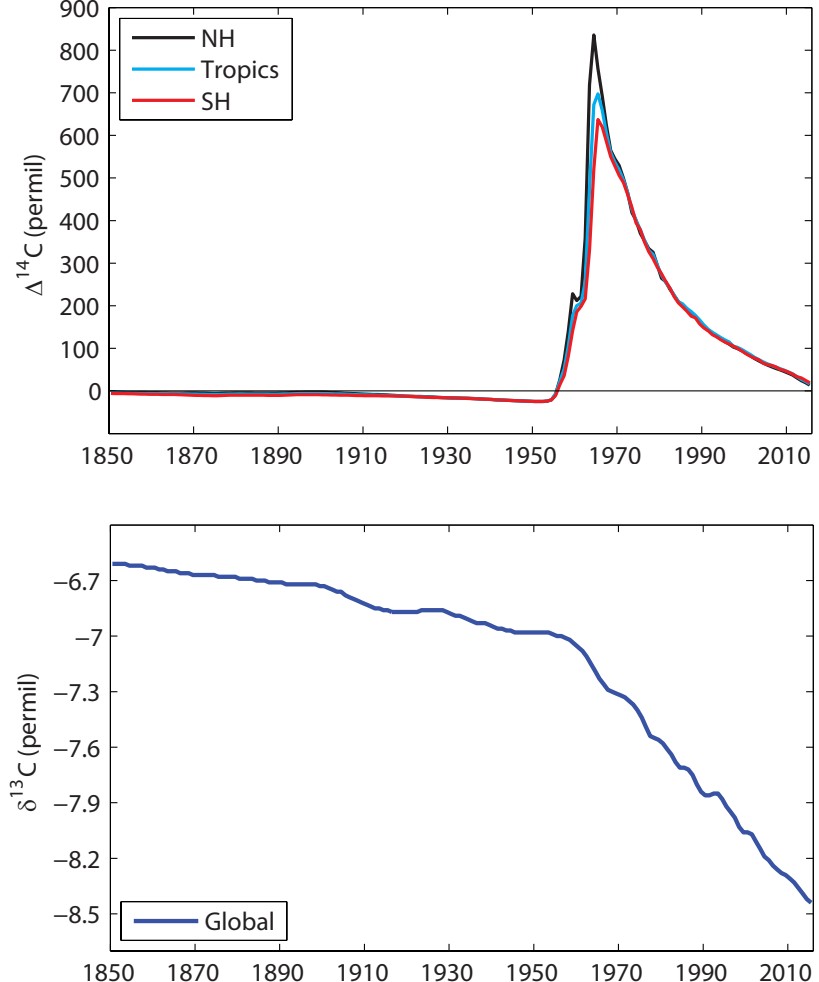

**Figure 1.** Historical atmospheric forcing datasets for $\Delta^{14}C$ in $CO_2$ (top) and $\delta^{13}C$ in $CO_2$ (bottom) compiled for CMIP6. Annual mean values of $\Delta^{14}C$ are provided for three zonal bands representing the Northern Hemisphere (30°N-90°N), the Tropics (30°S-30°N) and the Southern Hemisphere (30°S-90°S). Annual mean, global mean values are provided for $\delta^{13}C$. Tabulated data are provided in Table S1.



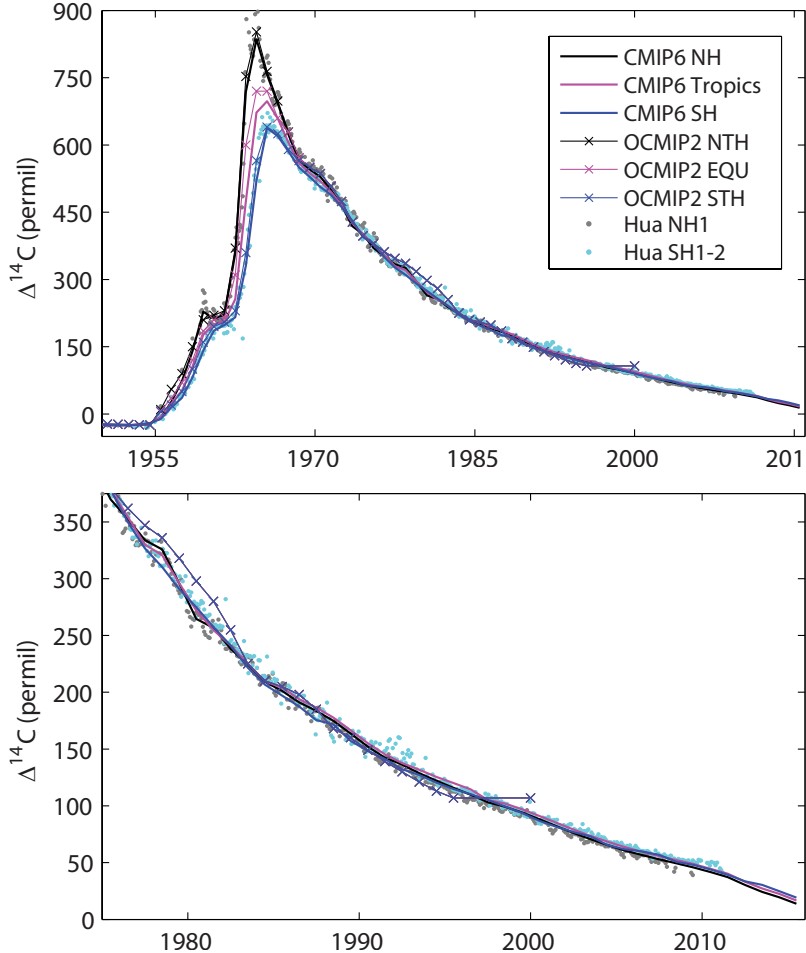

**Figure 2.** The Δ¹⁴C atmospheric forcing data for CMIP6 compared to the forcing data used in OCMIP2 and the Northern and Southern Hemisphere tree-ring and atmospheric data compilations of Hua et al. (2013). Zones NH1 and SH1-2 from Hua et al. (2013) are used, to correspond to the Northern Hemisphere north of 30°N and the Southern Hemisphere south of 30°S. Tropical data from Hua et al. (2013) is not shown, for clarity.



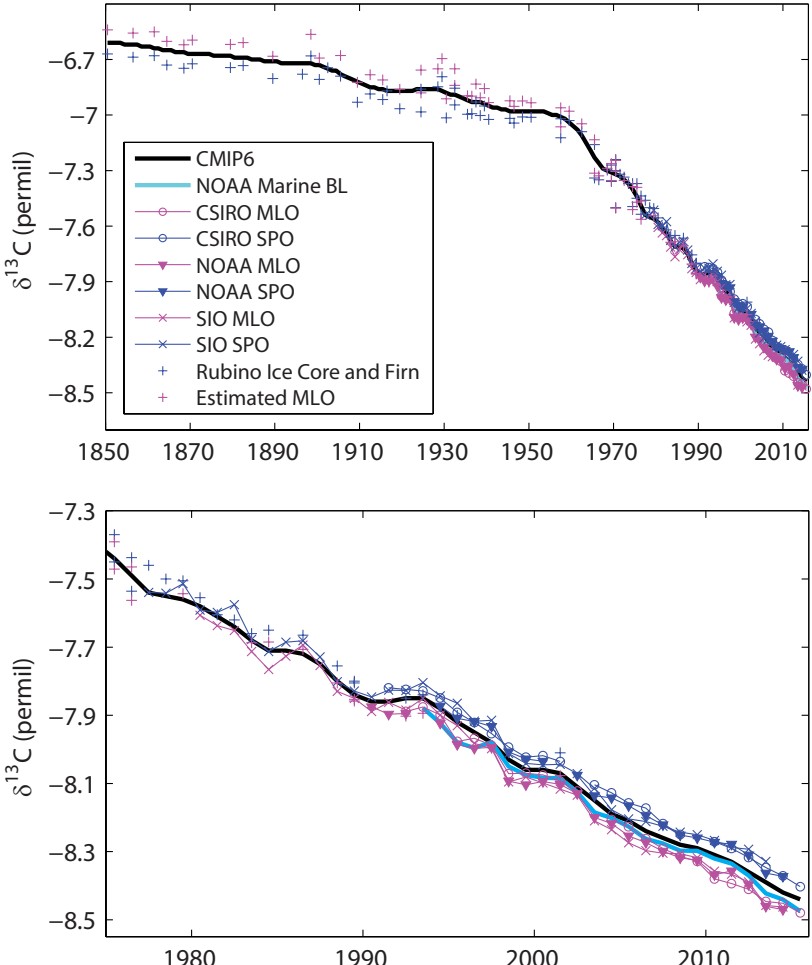

**Figure 3.** Observations of δ¹³C and the δ¹³C atmospheric forcing data for CMIP6 over the full time period 1850-2015 (top) and over the recent period 1975-2015 (bottom). The δ¹³C atmospheric forcing data for CMIP6 is shown in black, as in Fig. 1. Data from SPO and ice core and firn samples are shown in blue, and data from MLO and estimated data for MLO based on ice core and firn samples and the regression from Keeling et al. (2011) are shown in purple. Measurements conducted at different laboratories are shown with different symbols. Data from NOAA and SIO have been adjusted with their average laboratory offset from CSIRO. The global mean δ¹³C estimated by NOAA based on a larger network of flask sampling stations over 1993-2015 is shown in light blue (NOAA Marine Boundary Layer).