# Peer review of "Compiled records of carbon isotopes in atmospheric CO2 for historical simulations in CMIP6"

_Geoscientific Model Development, 2017_

## Short Comment (SC1) · 26 Jul 2017

Dear authors,

please include a DOI (or the ESGF equivalent identifier) for the data in the Data Availability Section.

Best,

Astrid Kerkweg (Executive Editor)
* * *

---

## Referee Comment (RC1) · K. Currie (Referee) · 4 Aug 2017

This manuscript describes the compilation of global data records of atmospheric 14CO2 and 13CO2 for use in modelling studies, in particular CMIP6. The methods used for compiling and standardising the individual data sets is well described and justified with appropriate caveats given.

The resulting data compilations are provided both as tables in the supplementary section, and via a website link.

The data compilations themselves will be useful, not only for CMIP6 but also for other modelling studies, and this manuscript is a valuable addition to the metadata as well as standing as a paper in its own right.

The manuscript is well written, is easy to read and the diagrams and tables are clear. The limitations and caveats for use of the data compilations are succinct and well laid out.

Even the descriptions of the standardising of the data sets, which are necessary, but can be tedious to read, are brief and to the point without loss of clarity.

I have only few comments, detailed below:

Page 2, line 14 – the atmospheric exchange with both the terrestrial and oceanic biospheres is on annual as well as decadal timescales, as evidenced by annual signals of 14C in tree rings and corals.

Figure 2 – caption to include explanation of the two panels, as is done in Figure 3.

---

## Short Comment (SC2) · 13 Sep 2017

The paper present a compilation of atmospheric $\Delta$14C and $\delta$13C for the period 1850-2015 for use in the Coupled Model Inter-comparison Project 6 (CMIP6) for models simulating carbon isotopes in their ocean or terrestrial biosphere models. The compiled data not only are important for CMIP6 model simulations but can also be used for other modelling studies. The paper is well written and structured and the methods are well described. However, there are several points, which need to be improved. I therefore recommend the paper should be acceptable for publication after addressing the below minor issues.

1. For Southern Hemisphere $\Delta$14C, • The Wellington data from 1955-1983 are

[Figure]

used for the compilation. Why the Wellington data from 1984-2014 are not used for the compilation given the Wellington-Heidelberg offset is known (on average 4‰ higher for the Wellington data as stated in p.5)? • Please cite references for recent data from Macquarie Island (2007-2009 & 2011), Cape Grim (after 2008) and Neumeyer (after Jan 2008). In Levin et al. (2010), data from Macquarie Island and Neumeyer were reported for the period Dec 1992 – Feb 2004, Apr 1987 – Dec 2008 and Feb 1983 – Jan 2008, respectively.

2. For Northern Hemisphere $\Delta$14C, Izana at $\sim$28oN do not belong to the NH zone (30oN-90oN) defined by the paper. Therefore these data cannot be employed for the compilation of the NH zone. Instead, these data can be used for the Tropics (30oS-30oN).

3. For tropical $\Delta$14C, • Why the data from Mauna Loa & Kumukahi (Hawaii) and Samoa (2001-2007) are not used for the compilation given the lab offset is known for the recent period (within 2-3‰ as stated in p.6)? • A brief description of the model for estimation of tropical $\Delta$14C should be presented, so the readers don't have to read Naegler and Levin (2006) in order to follow the current paper. What are the parameters for air exchange between the 4 atmospheric boxes? Is the atmosphere (4 boxes) a portion of a carbon cycle model?

4. I think the authors should add another Table to give a summary on what atmospheric 14C records during what time are used for the compilation. This will make easier for the readers to follow the paper.

5. p.6, "The version we describe here (version 2) incorporates new and updated $\Delta$14C data from Heidelberg University, . . .". Again, the authors should give references for the more recent Heidelberg data. If they have not been published yet, please mention "unpublished data".

6. p.2, "Records of atmospheric $\Delta$14C and $\delta$13C have been extended into the past using measurements of cellulose from tree rings and of CO2 in air from ice sheets

(ice cores and firn) respectively, . . .". Instead of "cellulose from tree rings" the authors should mention only "tree rings". It is because some IntCal13 and SHCal13 raw data are from tree rings with Acid-Base-Acid treatment and not cellulose or hemicellulose extracted from tree rings.

7. p.8, for atmospheric $\delta$13C, "The revised procedure ensures that all corrections are consistently applied to all samples measured at CSIRO since 1990, including all ice-core, firn air and flask measurements. We do not include measurements conducted at NOAA reported in Rubino et al. (2013)." Please mention why measurements conducted at NOAA reported in Rubino et al. (2013) are not used for the compilation.

---

## Author Comment (AC1) · 13 Oct 2017

We appreciate the comments from the reviewers, which have improved the manuscript. Below we detail revisions we have made to the manuscript to address each comment, and provide further information in response to some comments.

SC1 from A. Kerkweg

Please include a DOI (or the ESGF equivalent identifier) for the data in the Data Availability Section.

We added citations to the datasets at input4mips/ESGF including doi's (Graven et al. 2017ab).

RC1 from K. Currie

This manuscript describes the compilation of global data records of atmospheric 14CO2 and 13CO2 for use in modelling studies, in particular CMIP6. The methods used for compiling and standardising the individual data sets is well described and justified with appropriate caveats given.

The resulting data compilations are provided both as tables in the supplementary section, and via a website link.

The data compilations themselves will be useful, not only for CMIP6 but also for other modelling studies, and this manuscript is a valuable addition to the metadata as well as standing as a paper in its own right.

The manuscript is well written, is easy to read and the diagrams and tables are clear. The limitations and caveats for use of the data compilations are succinct and well laid out.

Even the descriptions of the standardising of the data sets, which are necessary, but can be tedious to read, are brief and to the point without loss of clarity.

I have only few comments, detailed below:

Page 2, line 14 – the atmospheric exchange with both the terrestrial and oceanic biospheres is on annual as well as decadal timescales, as evidenced by annual signals of 14C in tree rings and corals.

Replaced "decadal timescales" with "annual to decadal timescales"

Figure 2 – caption to include explanation of the two panels, as is done in Figure 3.

Added explanation of the two panels: "over the time period 1950-2015 (top) and over the recent period 1975-2015 (bottom)."

SC2 from Q. Hua

The paper present a compilation of atmospheric Δ14C and δ13C for the period 1850-2015 for use in the Coupled Model Inter-comparison Project 6 (CMIP6) for models simulating carbon isotopes in their ocean or terrestrial biosphere models. The compiled data not only are important for CMIP6 model simulations but can also be used for other modelling studies. The paper is well written and structured and the methods are well described. However, there are several points, which need to be improved. I therefore recommend the paper should be acceptable for publication after addressing the below minor issues.

1. For Southern Hemisphere Δ14C, The Wellington data from 1955-1983 are used for the compilation. Why the Wellington data from 1984-2014 are not used for the compilation given the Wellington-Heidelberg offset is known (on average 4‰ higher for the Wellington data as stated in p.5)?

We appreciate this comment and other comments suggesting that additional available data should be included in the compilation. Our aim here was to provide a consistent dataset over the historical period that is easy for modellers to use, and we focused on previous work by I. Levin and her colleagues who manage the only currently running, long-term global network for $\Delta^{14}CO_2$. We agree that the compilation would be improved by including more data, and we hope to pursue this in future versions, conducting some of the data comparisons that are needed to ensure consistency but that were not feasible for this study. However, we do not expect that incorporation of further datasets will substantially change annual $\Delta^{14}CO_2$ for zonal bands that we report in this compilation. We have added some text in section 3 and in the discussion about laboratory comparisons enabling improved data compilations in the future.

The Wellington-Heidelberg laboratory offset was characterized for the data prior to 1984. A recent laboratory intercomparison run by the Heidelberg lab in 2015 did include the Wellington lab (Hammer et al. 2016), but analysis of the Wellington-Heidelberg offset for the intervening decades was not done and therefore we are not confident that we can apply the same offset to the entire record. We do, however, hope to incorporate recent Wellington data in future versions of the data compilation after further data comparison. Recent work by Turnbull et al. ACPD 2017 demonstrate that recent differences between $\Delta^{14}CO_2$ measured at Cape Grim by the Heidelberg lab and $\Delta^{14}CO_2$ measured at Wellington by the Wellington lab are small. Therefore, inclusion of Wellington data is unlikely to substantially change the annual Southern Hemisphere $\Delta^{14}CO_2$ we report.

Please cite references for recent data from Macquarie Island (2007-2009 & 2011), Cape Grim (after 2008) and Neumeyer (after Jan 2008). In Levin et al. (2010), data from Macquarie Island and Neumeyer were reported for the period Dec 1992 – Feb 2004, Apr 1987 – Dec 2008 and Feb 1983 – Jan 2008, respectively.

We have added a citation to Levin et al. unpublished data. We also provide a link in Table 1 to the Heidelberg University data center where these data will be hosted.

2. For Northern Hemisphere Δ14C, Izana at ~28oN do not belong to the NH zone (30oN-90oN) defined by the paper. Therefore these data cannot be employed for the compilation of the NH zone. Instead, these data can be used for the Tropics (30oS-30oN).

We clarified that Izana data are not used alone to specify NH $\Delta^{14}CO_2$, but an offset is applied. "Even though Izaña is located at 28°N, slightly south of the 30°N bound, we use data from Izaña to specify Northern Hemisphere $\Delta^{14}C$ for the period 1985-88 when very little data are available, after correcting for the mean difference between Izaña and the average of Izaña, Jungfraujoch, and Alert over 1989-97." For 1989-1997, since Izana is so near to the 30°N bound, we take the approach of calculating an average for the northern stations for NH $\Delta^{14}CO_2$, and then applying an offset to calculate tropical $\Delta^{14}C$ based on measurements at Merida, located at 8°N. The comparisons we report with other data indicate the approach provides good results.

3. For tropical Δ14C, Why the data from Mauna Loa & Kumukahi (Hawaii) and Samoa (2001-2007) are not used for the compilation given the lab offset is known for the recent period (within 2-3‰ as stated in p.6)?

Since the Mauna Loa & Kumukahi (Hawaii) and Samoa data are only available for a few years, we did not use them explicitly in the compilation. Instead, we used the data to compare with the calculation of tropical $\Delta^{14}C$ using a fixed offset with Northern Hemisphere $\Delta^{14}C$. We do hope to update these records and incorporate them in future versions of the compiled records. We added a note that the current 2-3‰ comparability between laboratories, while similar to measurement uncertainty, is not as good as the current WMO target, and we hope that this paper helps to stimulate work on laboratory comparisons leading to further compilation of existing records.

A brief description of the model for estimation of tropical $\Delta14C$ should be presented, so the readers don't have to read Naegler and Levin (2006) in order to follow the current paper. What are the parameters for air exchange between the 4 atmospheric boxes? Is the atmosphere (4 boxes) a portion of a carbon cycle model?

We have added some description of the model, also fixing an error in the number of tropospheric boxes from the previous draft. "The atmospheric box model is part of a carbon cycle model which also simulates other atmospheric species and radioisotopes, and the model exchange parameters were optimized to match atmospheric data including $\Delta^{14}C$, $SF_6$ and $^{10}Be/^7Be$ (Naegler and Levin, 2006). The model includes six tropospheric boxes separated at the Equator, 30° and 60° in each hemisphere." We have not included the exchange parameters but we have added references to the Hesshaimer and Naegler PhD theses including these details.

4. I think the authors should add another Table to give a summary on what atmospheric $\Delta14C$ records during what time are used for the compilation. This will make easier for the readers to follow the paper.

We have added Table S3 to summarize what atmospheric $\Delta^{14}CO_2$ records during what time are used for the compilation.

5. p.6, "The version we describe here (version 2) incorporates new and updated $\Delta14C$ data from Heidelberg University, : : :". Again, the authors should give references for the more recent Heidelberg data. If they have not been published yet, please mention "unpublished data".

We have added citations to Hammer and Levin 2017 and to Levin et al. unpublished data here.

6. p.2, "Records of atmospheric $\Delta14C$ and $\delta13C$ have been extended into the past using measurements of cellulose from tree rings and of CO2 in air from ice sheets (ice cores and firn) respectively, : : :". Instead of "cellulose from tree rings" the authors should mention only "tree rings". It is because some IntCal13 and SHCal13 raw data are from tree rings with Acid-Base-Acid treatment and not cellulose or hemicellulose extracted from tree rings.

Thank you for this correction, we have removed "cellulose from" from this statement.

7. p.8, for atmospheric $\delta13C$, "The revised procedure ensures that all corrections are consistently applied to all samples measured at CSIRO since 1990, including all ice-core, firn air and flask measurements. We do not include measurements conducted at NOAA reported in Rubino et al. (2013)." Please mention why measurements conducted at NOAA reported in Rubino et al. (2013) are not used for the compilation.

We added a statement that the NOAA measurements were not used because the NOAA-CSIRO laboratory offset at the time that these observations were made, relative to the updated CSIRO calibration, could not easily be identified. However, these data from NOAA only comprise 24

individual firn air samples, in comparison to 154 individual samples measured at CSIRO from ice cores and firn, and only 3 of the NOAA samples are from the period prior to the flask data.